# Design considerations for technology-assisted fall-resisting skills training trials in older adults: A pilot and feasibility study

Elisabeth G. van der Hulst[1,2]*, Kenneth Meijer[1], Pieter Meyns[2‡],
Christopher McCrum[1‡]*

1 Department of Nutrition and Movement Sciences, NUTRIM Institute of Nutrition and Translational Research in Metabolism, Maastricht University, Maastricht, The Netherlands, 2 REVAL Rehabilitation Research Center, Hasselt University, Diepenbeek, Belgium

‡ PM and CM are joint senior authors on this work.
* liset.vanderhulst@maastrichtuniversity.nl (LvdH); chris.mccrum@maastrichtuniversity.nl (CM)

## Abstract

Training fall-resisting skills can prevent falls in older adults. These fall-resisting skills include proactive gait adaptability, gait robustness, and reactive gait recovery, which allow people to effectively avoid, resist, and recover from balance threats, respectively. This pilot study guided the design of an RCT of fall-resisting skills training by investigating key design factors, such as the design of a placebo-control group, obstacle difficulty settings, exploring evaluation methods for gait robustness, testing the effect of task unpredictability on anxiety, and the general feasibility. Eleven healthy older adults performed non-task-specific "placebo" balance tasks and assessment and training tasks for each fall-resisting skill. Placebo tasks included static weight-shifting exercises and dual-task walking. For the fall-resisting skill tasks, participants walked on a treadmill under different conditions. For proactive gait adaptability, participants avoided projected obstacles varying in size, approach speed, and available response time. Gait robustness was assessed using perturbations of increasing magnitude, where the margin of stability following each perturbation was compared with participants' perceived balance loss and researchers' observations. For reactive gait recovery, perturbations with increasing unpredictability were applied, after which participants reported their anxiety scores. Weight-shifting tasks were perceived as balance training by most participants, indicating their potential as placebo tasks. Obstacle avoidance difficulty increased most with fast approach speed and large obstacle sizes. A margin of stability-based threshold did not consistently align with perceived balance loss or observer judgement. Anxiety did not increase with more unpredictable perturbation tasks when introduced gradually. Fall-resisting skill tasks generally were feasible for older adults.

**Data availability statement:** Data related to this study can be found at the OSF project page at https://osf.io/rg7kq/ Further information can be obtained by contacting the corresponding authors.

**Funding:** This work was supported by the special research fund 2022 call for doctoral grants in the framework of BOF UHasselt – Maastricht University cooperation [BOF22DOCUM13]. The funders had no role in study design, data collection and analysis, decision to publish, or preparation of the manuscript.

**Competing interests:** I have read the journal's policy and the authors of this manuscript have the following competing interests: Kenneth Meijer serves as a scientific advisory board member for DIH Medical and expenses related to this activity are paid by DIH, whose Motek devices are used in this research. This relationship did not influence the research, and DIH had no involvement in the conception, conduct, or reporting of this study. The remaining authors have no conflicts of interest to declare.

## Introduction

Falls among community-dwelling older adults occur most frequently during ambulation due to mechanical disturbances such as trips, slips and stumbles [1–7]. As evidenced by the increasing burden that falls place on individuals and healthcare systems in our ageing societies [8–12], current approaches to fall prevention are falling short. Given that one of the strongest predictors of falling is a previous fall [13], effective primary prevention strategies could play a significant role in overall fall prevention.

There is strong evidence that exercise is effective in reducing falls [14,15] with increasing effectiveness when challenging balance training [14,15], gait adaptability training [16] or perturbation-based balance training [16,17] are implemented. Exercise interventions that require a longer period of adherence to see an effect seem to be more difficult to implement in practice at the necessary levels of dose and adherence [18,19], particularly in healthy older adults who do not necessarily yet feel, recognise or admit the need to participate in fall prevention interventions [20]. A recent online survey of European healthcare professionals also found that three of the five main barriers related to implementing fall prevention were time-related [21]. Therefore, more task-specific approaches that do not require such sustained adherence and are time-efficient for the participant may be attractive options.

Looking more closely at the phases of such an obstacle-induced fall, we can identify specific fall-resisting skills that can help older adults avoid such falls, as outlined previously [22,23]: 1) Proactive Gait Adaptability (PGA): the ability of a person to detect and adapt their gait to avoid a potential threat; 2) Gait Robustness (GR): the ability of a person to resist disturbances without losing stability; and 3) if stability is lost, the last line of defence is Reactive Gait Recovery (RGR): the ability of a person to react to, respond to, and recover from stability loss once it occurs. Here, we use balance to refer to the collection of processes and abilities (including sensory, motor, and cognitive components) that control the body's position and motion to maintain stability within the current movement context. Stability in the current movement context is defined as maintaining a centre-of-mass (COM) and base-of-support (BOS) relationship that allows successful performance of the task (i.e., walking (without falling)). Instability occurs when the COM–BOS relationship prevents successful task performance, either at a single instant or over a period of time. To better understand how each of these fall-resisting skills might contribute to reducing fall risk in older adults, well-designed trials are needed. However, several practical and methodological issues exist when designing such trials, as we will outline in the following sections.

Obstacle avoidance tasks are frequently studied as a method to evaluate and improve PGA in older adults [24–26]. It is important to establish the right difficulty level when designing assessments and interventions to prevent both ceiling effects, where tasks are too easy, limiting improvement potential, and floor effects, where tasks are too challenging, potentially causing anxiety, frustration or dropout. While traditional approaches involve physical obstacles, virtual or projected obstacles offer greater flexibility in manipulating training parameters such as obstacle size, shape

and colour, available response time, and obstacle approach speed. Decreased available response time negatively impacts obstacle avoidance performance [24,26]. However, the impact of the obstacle size, approach speed and the combination of these parameters on task difficulty, particularly when using virtual obstacles, remains underexplored.

Of the three fall-resisting skills addressed here, GR is the least established in the literature. As a result, methods for assessment and training are less developed. Based on our definition of GR as "the ability of a person to resist disturbances without losing stability", this implies that we need to be able to assess at what perturbation size or intensity a person loses stability during walking. For reactive balance during stance perturbations, the Stepping Threshold Test [27,28] is an example of a progressive, threshold-based test that can analyse robustness of standing stability to perturbations of increasing intensity. However, an equivalent for GR that could objectively determine when a person deviates from "steady state gait" due to overt stability loss does not yet exist, though a dynamic version of the Stepping Threshold Test using subjective judgement of stability loss has been reported [29]. In contrast to stance, the body configuration is frequently unstable during the normal gait cycle, making precise assessment of when stability is lost more complex during walking. In some previous studies, the number of recovery steps to return to baseline walking stability was evaluated as the number of steps to return to within 0.05m of the mean margin of stability of the 10 steps prior to the perturbation [30,31]. The group later updated this to a margin of stability value three standard deviations or greater away from the mean value since an individualised method was preferable compared to an absolute threshold due to individual differences in the variability of steady state gait [32]. However, neither the subjective judgement nor the three standard deviations approach was formally evaluated or compared to any other threshold.

While perturbation-based balance training (PBT) is an effective method to improve RGR in older adults [33,34], the perturbations, particularly their necessary unpredictable nature [35], can induce anxiety, potentially increasing dropout rates, lowering adherence, and reducing the effectiveness of the training [17,36]. Okubo, et al. [36] specifically showed that introducing different types and multiple possible perturbation locations on a walkway increased anxiety and dropout. On the other hand, Wong, et al. [37] did not find an effect of perturbation intensity unpredictability during multi-direction surface translation perturbations during standing on stressfulness and electrodermal responses. A recent review made an explicit recommendation that there is a need to "Determine strategies to alleviate anxiety in participants undertaking PBT to ensure clinical feasibility" [33]. Until now, little is known about how the introduction of multiple perturbation types and increasing difficulty during treadmill walking affects anxiety in older adults.

In trials that assess fall-resisting skills, factors such as older adults' enjoyment, the clarity of instructions, and perceived task difficulty can play an important role in feasibility and participant adherence. While two studies have reported qualitative feedback from older adults specifically on PBT [38,39], showing it to be generally well accepted, broader insights into how older adults experience different types of fall-resisting skills assessment and training tasks remain limited. To ensure acceptability and manageability in trials of fall-resisting skills, it is important to evaluate these factors across various types of assessment and training tasks.

A final methodological issue in intervention studies targeting fall-resisting skills is the design of an appropriate control condition. Without a well-matched placebo control condition, it becomes difficult to isolate the effects of the specific training mechanisms of interest from improvements driven by other potential mechanisms and non-specific factors [40] such as visual [41] or cognitive [42] stimulation, and increased confidence or reduced anxiety [43,44]. In the case of training fall-resisting skills, concern about falling can be reduced by exercise [44] and concern about falling can influence stepping performance [45]. Despite this, studies on fall-resisting skills training often lack control interventions that align with current recommendations for the design of placebo-controlled groups [40], namely that placebo interventions should replicate all components of the active training, such as duration, setting, and therapist interaction, except for the specific training mechanisms. In addition, the control intervention must be credible as a treatment and evoke expectations of improvement like those of the experimental intervention [40].

To address these gaps and inform the design of a future fall-resisting skills training trial, we aimed to: 1) Check if selected non-task-specific balance tasks can be perceived as training and potentially used as placebo control tasks; 2) Examine how different combinations of obstacle parameters affect obstacle crossing difficulty for older adults; 3) Explore evaluation methods for determining a stability loss threshold for GR assessment; 4) Investigate how increasing perturbation task unpredictability corresponds to older adults' anxiety; and 5) Check the feasibility of various fall-resisting skill tasks, specifically focusing on instructions, anxiety, enjoyment, and difficulty. For aim 2, we expect that larger obstacles, shorter available response times, and faster approach speeds increase obstacle crossing challenge. Smaller obstacles with longer available response times moving at the speed of the treadmill are expected to be the easiest combination. For aim 4, we expect that with increasing task unpredictability, anxiety in older adults will increase, but not to such a high level that the task will not be feasible anymore. There are no specific hypotheses formulated for aims 1, 3, and 5 since these aims are exploratory to assess the practicality and feasibility of the tasks.

## Methods

### Study design, setting, and participants

This pilot and feasibility study was conducted in preparation for a multicentre trial in Hasselt and Maastricht Universities. Ten participants were planned for this pilot study. Healthy community-dwelling older adults were recruited through posters and flyers in hospitals, universities, community centres, and fitness centres around Maastricht (Netherlands) and Hasselt (Belgium). Recruitment started and ended in Maastricht on May 1st 2024 and July 22nd 2024, respectively and started and ended in Hasselt on November 25th 2024 and January 22nd 2025, respectively.

Participants were eligible if they were aged 65 years or older, with no self-reported health conditions affecting balance or walking, and no recent history of lower limb injury. Exclusion criteria included any self-reported neurological, sensory, neuropathic, vestibular, or musculoskeletal disorders, as well as any other conditions affecting balance or walking. This roughly aligned with our intended target population for our future trial (see inclusion criteria on the trial registration page: https://onderzoekmetmensen.nl/en/trial/57562).

Overall, we consider a sample of ten older adults sufficient for the aims of this pilot study. For aim 2, we conducted a sensitivity power calculation. With ten participants (one group) completing 18 repeated measures across different task conditions, a repeated measures ANOVA with task factor combination as the single factor with power 0.8 for an effect size $f = 0.15$. Additionally, a Wilcoxon signed-rank test for the easiest and hardest combination of factors can detect large effects ($d = 1$) with the same sample size and power. For feasibility outcomes, both qualitative and quantitative, we consider ten participants to be sufficient, supported by our research group's previous experience.

Participants provided written informed consent prior to participating. The study was approved by the FHML Research Ethics Committee of Maastricht University (FHML-REC/2024/034) and the Medical Ethics Committee of Hasselt University (B1152024000019) and was performed according to the Declaration of Helsinki.

### Experimental setup

Measurements were performed using the Computer Assisted Rehabilitation Environment (CAREN; Motekforce Link, Amsterdam, The Netherlands) in Maastricht and the Gait Real-time Analysis Interactive Laboratory (GRAIL; Motekforce Link, Amsterdam, The Netherlands) in Hasselt, which include a dual-belt force plate-instrumented split-belt treadmill (1000 Hz), a 12 and 10-camera motion capture system, respectively (100 Hz; Vicon Motion Systems, Oxford, UK), three 2D video cameras, and a virtual environment providing optic flow. Six retroreflective markers were attached to anatomical landmarks (C7, sacrum, left and right trochanter, and left and right hallux). For the two weight shifting tasks (boat, city ride), these six markers were covered and two shoulder markers were added to control the application. Both devices are operated via the DFlow software of Motek and share most components. However, they differ in movement capability. The

CAREN includes a movable platform that can move in 6 degrees of freedom, whereas the GRAIL can only move in the pitch direction and make limited mediolateral movements. A safety harness was always used to protect the participant from falling in case of stability loss. Prior to starting the measurements, participants' height, weight, leg length and foot length were measured. The same researcher conducted all measurements.

## Experimental procedure

Participants started familiarisation trials by walking at 1 m/s for two minutes, after which the speed increased by 0.2 m/s every two minutes up to 1.6 m/s. These trials were recorded and during a rest period following the familiarisation, participants' individual stability-normalised walking speed [46] was calculated (see calculation procedure in "Data processing" section below) and used for all following walking tasks to ensure that all participants walked at approximately similar stability (i.e., MoS=~0.05m) during all tasks, which is not guaranteed when self-selected walking speeds are used [46].

Participants then started with non-task-specific balance tasks. The first task was a standing weight-shifting game in which they steered a boat through an obstacle course presented on the screen, slaloming around buoys as quickly as possible. This was followed by a similar weight-shifting task, steering a car through a city while avoiding upcoming vehicles. Both tasks are provided as standard applications on the CAREN and GRAIL systems (Motek Medical BV [47], as previously used in Sessoms, et al. [48] and Markham, et al. [49]). The boat and car avatars were both controlled via the tracked motion of reflective markers placed on the shoulders. The final non-task-specific balance task was walking for four minutes at the stability-normalised walking speed while performing a cognitive dual task: the Auditory Stroop Task [50]. Participants heard the words 'high' and 'low' spoken in either a high or low pitch through a one-ear headset and were instructed to name the pitch of the voice, ignoring the word's content. This task was used since in our future trial, training of the fall-resisting skills will increase in difficulty across sessions, in part due to the addition of this cognitive dual task. To ensure that the cognitive stimulation is similar for participants in the control condition, we will need to use this cognitive dual task in the control group as well.

After a short break, participants continued with the PGA tasks. The first PGA task was an obstacle avoidance task, to inform the design of the PGA assessment and training tasks in our future trial. White obstacles were projected onto the treadmill belt, each appearing at the predicted foot touchdown location. The expected foot touchdown location was estimated based on the mean stride time of the last five strides and the current position of the feet, determined using the toe markers. Three different parameters and their combinations were tested: obstacle size (0.5, 1, and 1.5 times the participants' foot length), approach speed (0.5, 1, and 1.5 times the treadmill speed), and available response time (1 and 2 strides before obstacle appearance). We selected these ranges of obstacle sizes, approach speeds, and response times to explore which factors most strongly affect obstacle avoidance difficulty. All combinations were tested with six obstacles per combination, resulting in 108 obstacles in total. The order of combinations was randomised across participants. In the next task, planned as a potential training task in the future trial, participants walked on the treadmill while aiming to step on white stepping stones projected onto the treadmill, avoiding those that unpredictably turned red. The final PGA task also involved projected stepping stones, but some were displaced 7.5 cm medially, requiring medial foot placement adjustments. Both stepping stone tasks contained seven obstacles or location shifts per minute, with a response distance of 1.4 meters.

To investigate potential evaluation methods for GR, the next task involved participants walking on the treadmill while unannounced treadmill acceleration (3m/s²) perturbations were delivered. Similar to previous work from our group [30–32,51–53], the acceleration started when the hallux marker of the to-be-perturbed limb passed the hallux marker of the opposite foot in the sagittal plane, meaning that when the foot touched down onto the treadmill belt, the belt was already moving at a greater speed, perturbing the limb for the duration of the stance phase. The belt decelerated again at toe-off of the perturbed limb. The perturbations started very small, with accelerations to a belt speed 105% of the walking speed, and subsequent perturbations gradually increased in steps of 5% to a maximum of 170% of the walking speed. Each leg was perturbed once for each percentage

increase in belt speed. After each perturbation, the participant was verbally asked the following questions: 1) Did you feel like you had to change your steps? 2) How stable did you feel on a scale of 1–7, with 1 being not stable at all and 7 being very stable. Following this, two potential GR training tasks were conducted. During the first GR training task, participants walked on the treadmill while the treadmill speed was pseudo-randomly changing, according to the following sine waves:

$$Treadmill\ speed\ sway\ = v_{tm} + ((0.5 \cdot sin(0.5 \cdot 2\pi \cdot time) + 0.8 \cdot sin(0.7 \cdot 2\pi \cdot time) + 0.7 \cdot sin(0.9 \cdot 2\pi \cdot time)$$
$$+\ 0.5 \cdot sin(1.5 \cdot 2\pi \cdot time)) \cdot \left(\frac{0.40775}{3.355}\right)$$

For the second GR task, participants walked on the treadmill while the treadmill platform moved pseudo-randomly mediolaterally according to the following formula, which was a modified version of the formula used in Zhu, et al. [54], based on the original formula presented by McAndrew, et al. [55]:

$$ML\ sway\ = \frac{0.083875}{3.355} * 0.5sin(0.16 * 2\pi * time) + 0.8sin(0.21 * 2\pi * time) + .7sin(0.24 * 2\pi * time)$$
$$+\ 0.5sin(0.49 * 2\pi * time)$$

Following the GR tasks and a short break, participants continued with the RGR tasks. Participants were informed they would complete a walking balance challenge but were not given any other details about the task. For each RGR task, participants experienced 12 perturbations (6 per leg) in a randomized order, applied approximately every 20 seconds. The first task included treadmill belt acceleration perturbations as described in the GR task description above, except that the belt accelerated to 180% of the walking speed [30–32,51–53]. Following this, participants received different tasks based on whether they were in Maastricht at the CAREN or in Hasselt at the GRAIL, due to slight differences in hardware. In Maastricht, participants were exposed to acute mediolateral (ML) perturbations induced by a platform shift. The displacement was determined by the following formula:

$$Displacement\ (m)\ =\ 0.025 + 4 \cdot v_{tm} \cdot 0.025$$

resulting in displacements ranging from 13.4 to 14 cm. Maximum speed ranged between 0.28 m/s and 0.29 m/s, platform accelerations between 0.98 m/s$^2$ and 1.03 m/s$^2$, and perturbation durations ranged from 1.58 seconds to 1.63 seconds. These perturbations were applied when the contralateral leg was in mid-stance. For the third task, AP and ML perturbations were combined in randomised order. For the final task, participants performed the Auditory Stroop Task while walking with combined AP and ML perturbations. In Hasselt, participants completed only one additional task with the treadmill belt accelerations while performing the Auditory Stroop Task.

Breaks were given after every block of tasks for each fall-resisting skill. More breaks were provided when requested. An overview of all tasks and their duration is shown in Table 1.

## Questionnaires

Falls incidence, cause, location, and consequences in the previous 12 months were assessed by a questionnaire based on the recommendations of Lamb, et al. [56] and Lord, et al. [57] (translated by McCrum [58]), that led with the question "In the past year, have you had any fall including a slip or trip in which you lost your balance and landed on the floor or ground or lower level?". Concern about falling was assessed using the translated version of the Falls Efficacy Scale International [59,60]. These were completed at the start of the session.

To gather information about the feasibility of the tasks, participants verbally answered open-ended questions about the task regarding the instructions, their experienced anxiety and stress during the task, and their enjoyment of the task, and rated the instructions, enjoyment, tension and anxiety on a 7-point Likert scale after every task. After each

**Table 1. Overview of the tasks performed.**

| Task category | Task | Duration (min) | Aim(s) |
|---|---|---|---|
| Familiarisation | Walking at different speeds | 8 | * |
| Non-task-specific balance tasks | Boat | 3 | 1, 5 |
| | Car | 3 | 1, 5 |
| | Walking with AST | 4 | 1, 5 |
| PGA | Obstacle avoidance | 12 | 2, 5 |
| | Stepping stones – obstacles | 3 | 5 |
| | Stepping stones – location shift | 3 | 5 |
| GR | Increasing acute AP perturbations | 10 | 3, 5 |
| | Continuous AP perturbations | 3 | 5 |
| | Continuous ML perturbations | 3 | 5 |
| RGR | AP perturbations (Maastricht + Hasselt) | 4 | 4, 5 |
| | ML perturbations (Maastricht) | 4 | 4, 5 |
| | AP + ML perturbations (Maastricht) | 4 | 4, 5 |
| | AP + ML perturbations + AST (Maastricht) | 4 | 4, 5 |
| | AP perturbations + AST (Hasselt) | 4 | 4, 5 |

* To determine stability normalised walking speed; PGA: Proactive Gait Adaptability; GR: Gait Robustness; RGR: Reactive Gait Recovery; AST: Auditory Stroop Test; AP: Anteroposterior; ML: Mediolateral. Aims: 1) are non-task-specific balance tasks potential control tasks; 2) obstacle parameters affecting difficulty; 3) determining a stability loss threshold for GR assessment; 4) effect of perturbation task unpredictability on anxiety; and 5) feasibility of various fall-resisting skill tasks.

non-task-specific balance task, participants answered questions about their expectations to improve on these tasks. After all non-task-specific balance tasks, participants were asked whether participating in such training (a combination of all non-task-specific balance tasks) could improve their balance and whether they would recommend such training to others.

## Data processing

For the stability-normalised walking speed, the mean anteroposterior margin of stability (AP MoS, Hof, et al. [61]) of the final 10 steps at foot touchdown at each familiarisation speed was used to determine the walking speed resulting in a MoS of 0.05m, based on the procedure described in McCrum, et al. [46]. For this and the GR task, MoS was calculated using the validated reduced marker set of Süptitz, et al. [62], specifically:

$$MoS = BoS - XcoM$$

where BoS was the location of the hallux marker of the anterior foot at foot touchdown and

$$XcoM = x + \frac{v_{x|T} + v_T}{\omega_0}$$

in which $x$ is the vertical projection of the CoM on the ground (estimated from the trochanter markers), $v_x$ is CoM velocity (m s$^{-1}$; measured as the average velocity of the trochanter and C7 markers), $v_T$ is treadmill belt speed, $\omega_0 = \sqrt{\frac{g}{l}}$ is the eigenfrequency of the pendulum, $g$ is the gravitational constant (m s$^{-2}$), and $l$ is effective pendulum length, measured as the standing leg length (greater trochanter to lateral malleolus), with all units of distance and length in meters. For the GR assessment task, 3SDs of the participant's 10 steps prior to each perturbation was calculated to determine the threshold. This individualised threshold was used to account for individual differences in MoS variability [32].

Data was processed in MATLAB 2024b. The marker positions were filtered with a low-pass second-order zero-phase Butterworth filter with a 12 Hz cut-off frequency. Force plate (50 N threshold) and foot marker data [63] methods were combined to determine foot touchdown and toe-off (as previously described in McCrum, et al. [64]). Obstacle hits were determined using the Center of Pressure (CoP) data recorded by the force plates. CoP signals were filtered using a second order low-pass Butterworth filter with a 15 Hz cut-off frequency. Obstacle position data was interpolated to the sampling frequency of the CoP. Stance phases were identified using a vertical ground reaction force threshold of 50N. The CoP trajectory was compared with the location of the moving obstacles projected onto the treadmill belt to identify whether the obstacle was hit or not. A hit was registered if the CoP position fell within an obstacle. For slow approach speeds (e.g., 0.5 times the treadmill speed), only the CoP positions immediately following foot placement were analysed.

### Aims assessment

For aim 1, we analysed participants' perceived ability to improve on the non-task-specific balance tasks, whether they believed training these tasks would improve their balance, and whether they would recommend the training to others by categorizing them as positive, neutral/don't know, or negative. The results are reported descriptively and with quotes from participants to present the different ways individuals described their experiences.

For aim 2, we calculated the sum of obstacle hits per combination of factors (size, speed, available response time) to assess the relative difficulty of each condition. Since the data were not normally distributed, we used a Friedman test to compare overall hit scores across different combinations of factors, followed by a Wilcoxon signed-rank test to compare the two combinations with the highest and lowest difficulty, excluding combinations that exhibited floor or ceiling effects. Visual inspection of the data distribution was also performed to support the interpretation of the results. All statistical analyses were conducted using SPSS (version 29.0.1.1), with the significance level set at $\alpha = .05$.

Aim 3 was assessed in the GR task with three potential methods to determine the stability loss threshold. Firstly, we checked if the MoS of the first step after the perturbation fell within three standard deviations of the mean MoS of the ten steps before the perturbation. Secondly, participants' responses after each perturbation of whether they had to change their steps (yes/no) and a score of 1–7 on their stability were collected. The third method included gait researchers (the authors) watching video recordings of the trials and evaluating whether an overt reactive gait response was observed. We descriptively and visually compared the outcomes of the three assessment methods (MoS, participant perception, and researcher evaluation), as well as the agreement between the gait researchers.

To address aim 4, we assessed participants' anxiety scores after each RGR trial with large mechanical perturbations. Scores were compared per trial, with each increasing in unpredictability. This was done separately for the participants measured in Maastricht and those in Hasselt, since the participants in Hasselt did not perform the ML perturbations.

For aim 5, each feasibility aspect – instructions, anxiety, enjoyment, and difficulty – was assessed by evaluating qualitative and quantitative criteria for each feasibility aspect. All criteria are shown in Table 2.

## Results

### Participants

Eleven older adults were recruited in this study. One participant dropped out during the obstacle avoidance task due to knee pain. Following discussion with the participant, it became clear that this was not caused by the task but was a pre-existing condition that the participant did not declare during screening. The data collected up to that point was included in the study for this participant. Recruitment was continued until we reached the initially planned ten participants who completed all tasks. Five participated in Maastricht, six in Hasselt. Four participants experienced a fall in the previous 12 months, with a mean of two falls. Participant characteristics are described in Table 3.

**Table 2. Criteria for feasibility of the tasks.**

| Domain | Quantitative | Qualitative | Criteria |
|---|---|---|---|
| Instructions | The number of misunderstandings or miscommunications | Questionnaire with open questions; Things that had to be mentioned to participants during the trial because of misunderstandings | If multiple misunderstandings within a person occur for multiple participants and multiple participants report difficulty understanding instructions, then the instructions may not be feasible to understand clearly and will need to be revised |
| Anxiety | The number of times participants ask to stop the task for anxiety, stress, or emotional reasons (if given) | Questionnaire with open questions and a scale after each task | If multiple people ask to stop during the task and/or multiple participants report very high anxiety/fear/discomfort during the tasks, the tasks may not be feasible in terms of anxiety for the target population |
| Enjoyment | | Questionnaire with open questions to gather information on what the older adults like, dislike, and enjoy about the examined tasks. | |
| Difficulty | Check the floor effect by checking the number of participants that cannot complete the task without harness assistance or stopping early due to imbalance and/or fatigue | | If multiple participants cannot complete the task without harness assistance or stopping due to instability/fatigue, then the tasks may not be feasible for the target population |

**Table 3. Participant characteristics.**

| Participants | Mean (SD) or n (%) (n = 11) |
|---|---|
| Age (years) | 73.5 (5.5) |
| Sex (M/F) | 6 (55)/5 (45) |
| Height (cm) | 165.7 (6.4) |
| Weight (kg) | 70.5 (13.8) |
| BMI (kg/m²) | 25.5 (3.9) |
| Normalized walking speed (m/s) | 1.08 (0.15) |
| FES-I score | 23.7 (13.78) |

## Missing data

Some participants had missing data for specific tasks. One participant, as described above, dropped out halfway through the obstacle avoidance task and for another there were technical issues, meaning that for both participants, only data from the first half of the task was included in the analysis. For the gait robustness task, one participant's data collection was stopped after perturbations with a 20% increase due to technical issues in the D-Flow application, data of one participant was not included in the analysis due to poor data quality and for three participants, no video data was collected due to technical errors. A detailed overview of missing data is shown in the supplementary material (Table 1 in S1 File).

## Aim 1: Non-specific balance tasks

Ten participants believed they could improve on the boat task, and one participant was not sure. Six participants believed they could improve on the city ride task, two were unsure, and three believed they could not improve. Five participants believed they could improve on walking with the dual task, three were unsure, and three did not believe so. One participant who was unsure about the walking with dual task mentioned the following:

*"I find it hard to say. It does not seem to have to do much with balance."*

For all three tasks combined as (hypothetical) training, two participants did not know whether following such a training could improve their balance, and nine participants believed it could. Open responses mentioned the following:

*"I think such a training would be very good for balance and reaction. All tasks are related to responding/reaction and attention".*

*"I think such a training would undoubtedly lead to improvements in balance, but the skills should be practiced in daily life."*

Ten participants would recommend such training to their peers, and one did not know. Some examples of the justifications for these scores were:

*"I would recommend this training to others since you're doing things you usually don't do."*

*"I would recommend this training because some people cannot balance very well and therefore quit/drop out of our walking club."*

Full responses can be found in the supplementary material.

### Aim 2: Effects of obstacle parameters on obstacle hits

The Friedman test revealed no significant effect of obstacle parameters settings on obstacle avoidance performance ($X^2(17) = 13.443$, $p = 0.706$). The easiest difficulty (mean rank of 5.21) was the combination of size = 1, speed = 1.5, and time = 2. The most difficult combination (mean rank of 11.86) had a size of 1.5, speed of 1.5, and time of 2. A Wilcoxon signed-rank test compared these two combinations and showed a significant difference (Z = 15, p = 0.038). No combination of obstacle parameters led to participants avoiding all obstacles or hitting all obstacles (no floor or ceiling effects). For every combination, at least one participant cleared all obstacles. Visual examination of the data (Fig 1) shows that

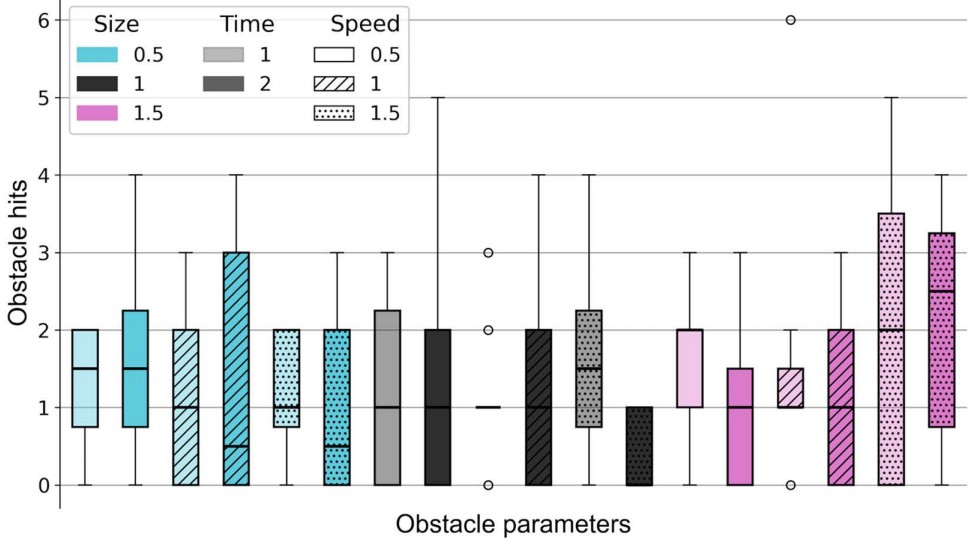

**Fig 1. Obstacle hits per combination of obstacle parameters.** Size is the size of the obstacle relative to the foot length. Time is the number of strides the obstacle appeared before it reached the subject. Speed is the approach speed of the obstacle relative to the treadmill speed. Boxplots show the median, interquartile range (25th – 75th percentile), whiskers (data within 1.5 x interquartile range) and outliers.

the combinations with the highest obstacle speed (1.5) and the biggest obstacle size (1.5) caused the most obstacle hits and, therefore, seem to be the most difficult combinations. The easiest combination was the middle size (1), fastest speed (1.5), and longest response time (2), with only three participants hitting one object each, while all others avoided all obstacles.

### Aim 3: Stability loss threshold

The MoS of the first recovery step after each perturbation showed inconsistent patterns across individuals (Fig 2). In theory, it would be expected that, as the percentage increase in belt speed during the perturbation increases, so too would the proportion of participants with their first recovery step being outside the 3SD MoS value. While sometimes occurring, this did not appear as a consistent pattern either within or across individuals.

   Across participants, the first perturbation with a step exceeding the 3SD threshold occurred between 15 and 35%. However, this was often followed by perturbations that did not lead to recovery steps outside the 3SD MoS threshold (Fig 2).

   Regarding the participants' perceptions, initially, participants answered the question 'Did you feel like you had to change your steps?' in a way that related more to whether they had felt something on the treadmill rather than whether they had to alter their stepping in response to the treadmill belt accelerations. Therefore, the question 'Did you feel something happening on the treadmill?' was added after the first four participants to make a clearer distinction between noticing a perturbation and needing to adjust one's steps. Overall, participants started reporting needing to adjust their steps between the 5 and 15% perturbations, with some individual variation (Fig 3). Notably, this 10% increase was also when participants typically first reported feeling something happening on the treadmill. Regarding the stability scores (On a scale of 1–7, how stable did you feel?), some participants reported a clear decrease in perceived stability as the percentage increase in belt speed during the perturbation increased, whereas others reported less pronounced changes, with stability scores remaining relatively consistent across intensities (Fig 3). The observations of the different gait researchers

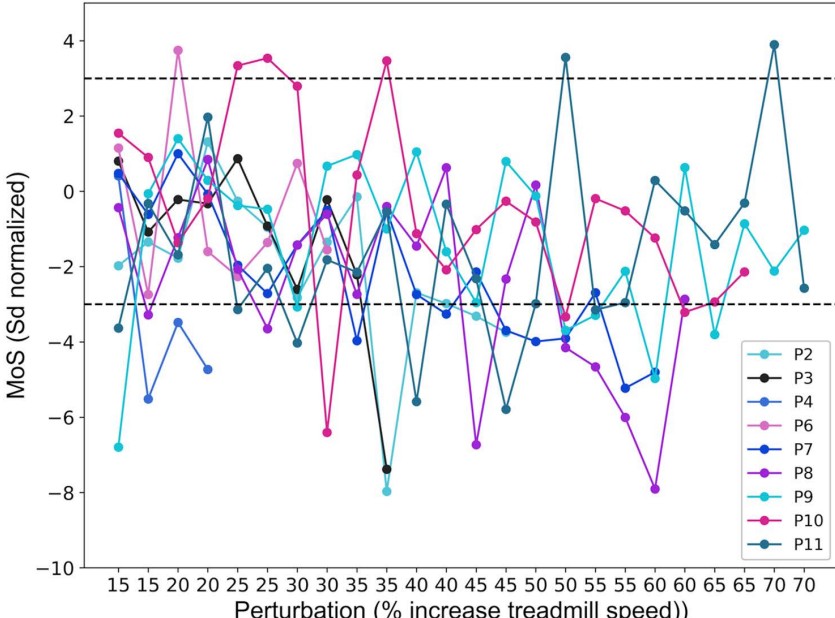

**Fig 2. SD normalised Margin of Stability during the first step after perturbations with increasing intensity.** Perturbations are applied twice for each intensity, once for each leg. MoS values are normalised in SD based on MoS during 10 steps of normal walking before each perturbation. Each line represents an individual participant with their data points connected. The black dashed lines indicate the ± 3 SD threshold.

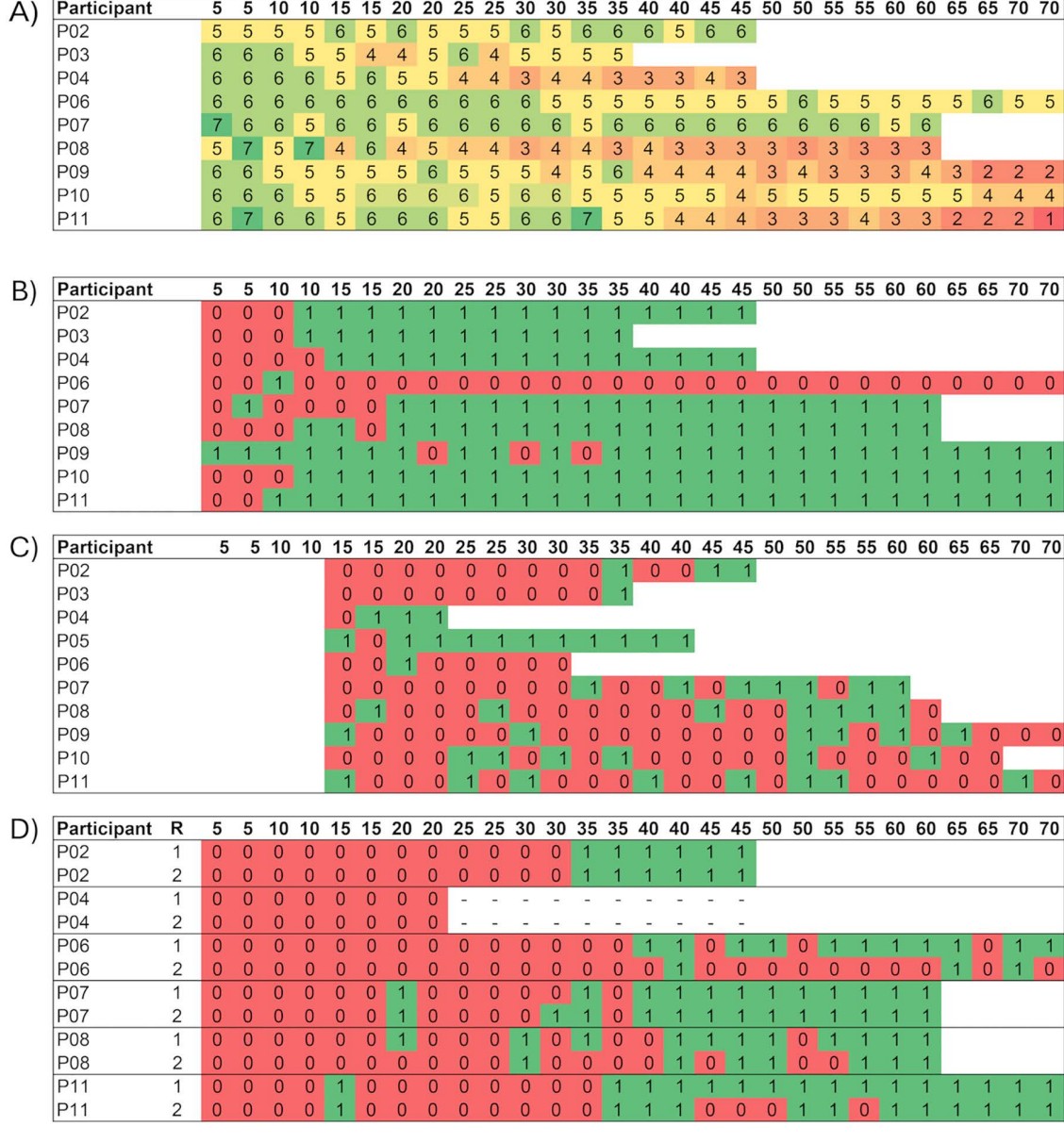

**Fig 3. A) Stability scores given by participants at each percentage increase in belt speed during the perturbation in response to the question "On a scale 1-7, how stable did you feel?" (1 = not stable at all, 7 = very stable). B)** Participant responses to the question "Did you feel like you had to change your steps?" (0 = No, 1 = Yes) after every perturbation. **C)** Margin of Stability values of the first step after each perturbation scored on whether they are inside or outside a 3 standard deviations threshold of the Margin of Stability values pre-perturbation (0=inside, 1=outside). **D)** Gait researcher observations of whether the participant had to change their steps (0 = No, 1 = Yes). R = Gait Researcher. *Note: Video data of P04 missing from 25% onwards.

had an agreement rate of 87%. The researchers mostly agreed upon the first occurrence of a recovery step. However, as the percentage increase in belt speed during the perturbation was raised beyond the occurrence of the first recovery step, some discrepancies occurred. These did not occur at the highest intensities, where both researchers consistently identified visible gait adjustments and deviations from normal gait, but rather at moderate intensities, where it was sometimes unclear whether a recovery step had occurred.

Regarding the correspondence of the three methods, participants' perceptions of stability did not consistently align with the 3SD MoS threshold. Many participants reported experiencing instability at lower perturbation intensities which caused only minor deviations in MoS. The gait researchers' observations corresponded slightly better to the threshold of 3SD MoS than the participants' perceptions did, but was still not entirely consistent.

### Aim 4: Anxiety-task unpredictability

In Maastricht, participants started with an anxiety score of 2.75±1.71 during the AP perturbations (Fig 4). As perturbation unpredictability increased, all participants' anxiety scores decreased from 1.33±0.58 during the trial with multiple perturbation directions to no anxiety at all (1.00±0) in the trial with multiple perturbation directions and a dual-task. All participants' anxiety scores decreased with the increasing unpredictability. One participant reported not experiencing any anxiety during all trials. Participants in Hasselt started with lower anxiety scores than in Maastricht (1.67±0.81). When a dual-task was added, one participant increased their anxiety score, while most others decreased their anxiety scores or did not report any anxiety during either trial (1.60±1.34).

### Aim 5: Feasibility

For the instructions, there were five minor misunderstandings. While walking with the AST, one participant did not say the words out loud initially. During the obstacle avoidance task it was not clear for one participant if they had to step over the obstacles or slalom around them. In the stepping stones task, it was unclear to one participant whether they were allowed to step beside the red obstacles when avoiding them or if they had to additionally only step on the white blocks. The task that had to be clarified most often for participants was the boat task: it was not clear to one participant that it was possible to slow down the boat by leaning or moving backwards; two participants needed extra instructions during the task to help them understand the boat's steering mechanism. One participant had two misunderstandings (boat task and AST), but the other misunderstandings all occurred in different participants. The score the participants gave for clarity of the instructions was high and similar for every type of task (Fig 5).

For the quantitative criteria of anxiety, no participants stopped any tasks due to anxiety, stress, or emotional reasons. Overall, participants reported low levels of anxiety (Fig 5). The experienced tension was a bit higher than their

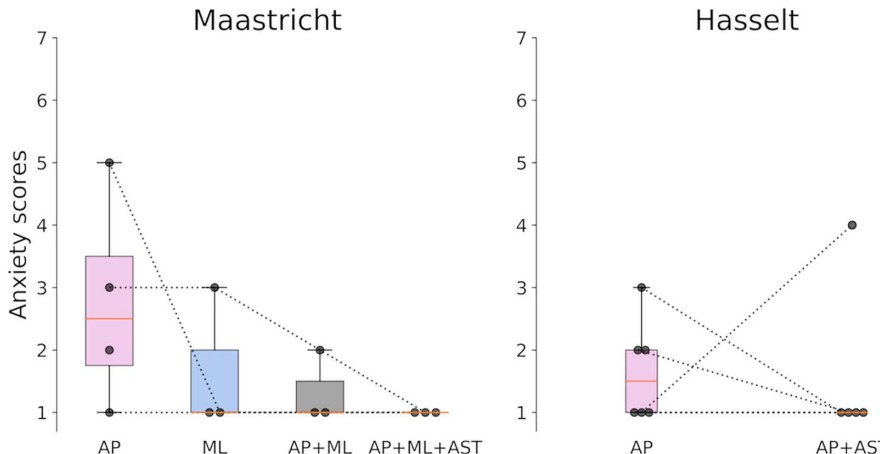

**Fig 4. Anxiety scores reported by participants for walking with perturbations with increasing unpredictability.** Scores are the answers to the question "On a scale from 1-7, how anxious did you feel during the task?", with a higher score indicating higher anxiety. Boxplots show the median, interquartile range (25th – 75th percentile), whiskers (data within 1.5 x interquartile range) and outliers. AP: Anteroposterior, ML: Mediolateral, AST: Auditory Stroop Test.

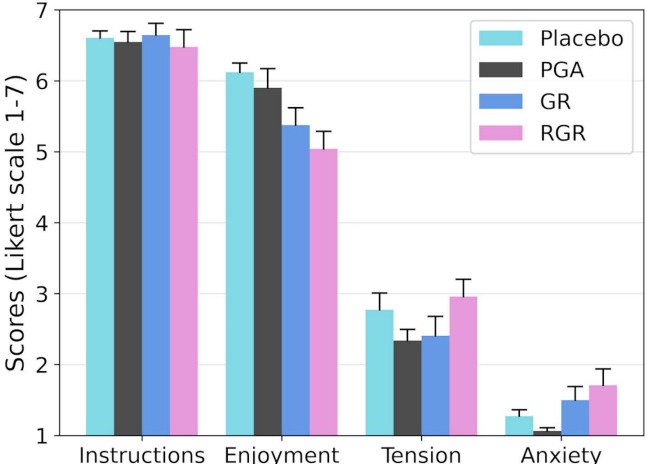

**Fig 5. Feasibility scores on instructions, enjoyment, tension, and anxiety experienced by the participants during the different task types reported on a Likert scale 1-7.** Note: higher scores reflect higher levels of perceived instruction clarity, enjoyment, tension, and anxiety. PGA: proactive gait adaptability; GR: gait robustness; RGR: reactive gait recovery.

experienced anxiety (Fig 5). Regarding participant feedback to the open questions, most participants experienced no stress or anxiety during the placebo tasks. Some felt tense because of the gaming aspect, or were nervous about giving the correct answers at the AST. For example, one participant said after the boat task:

*"I did not experience any stress or anxiety, but did have an adrenaline rush because of the game"*

The participants reported more nervousness during the city ride game compared to the boat game.

Most participants reported no stress or anxiety during the obstacle avoidance task. Five felt a bit nervous or tense, and one participant mentioned that this was specifically when the obstacles appeared faster. Two participants reported feeling a bit absent during the task. During the stepping stones tasks, some participants reported feeling a bit nervous, but the rest did not report any stress or anxiety.

During the GR tasks, most participants experienced no stress or anxiety. One participant was a bit nervous during the acute AP perturbation, and one was a bit scared to lose balance and didn't want to fall. Some participants felt slightly more stressed during the continuous AP perturbations. One participant also reported some anxiety during the continuous ML perturbations (platform sway) because she could feel that something was happening but could not place exactly what.

In the RGR tasks, the anxiety was a bit higher overall, but did not cause any participants to stop. Five participants mentioned they felt a bit nervous or anxious after the first RGR task with AP perturbations.

*"Sometimes I felt afraid to fall" (RGR AP)*

While the other six participants did not report any nervousness, anxiety or stress during this task.

*"I did not feel any nervousness, anxiety, or stress. You start to get used to it. I was curious when the perturbations would come." (RGR AP)*

Walking with the AST in combination with the perturbations made two participants more tense, and one reported this was because they did not want to make any mistakes in the AST. For more details on the experienced anxiety in the RGR tasks, see the results for aim 4 above.

Regarding enjoyment, the placebo tasks were rated as the most enjoyable and the RGR tasks the least enjoyable (Fig 5) though the difference in enjoyment was small (approximately only 1 point on Likert scale). Overall, participants described the placebo tasks as "fun", especially the boat game and the city ride. The AST was described as "good", "fun", "interesting", or "requiring concentration" by the participants. Most participants described the PGA tasks as "fun", "interesting", and "good". For example, one participant said:

*"The task was easy and fun. They are all nice tests and require attention and movement"* (Stepping stones moving blocks)

One participant described the obstacle avoidance task as "monotonous", and one participant mentioned that they did not like the PGA tasks because they had to look down constantly. Two participants specifically mentioned that the stepping stones task with the moving blocks was easy, while two others said it was difficult. Overall, the participants enjoyed the GR tasks less than the placebo and PGA tasks. The continuous AP perturbations evoked metaphorical descriptions: one person described it as being like an earthquake, another felt like a drunk person walking, and another participant reported feeling like the treadmill continuously bullied them. Similar to the PGA tasks, some participants reported the tasks as easy, while others found it more challenging. One participant described the GR task as follows:

*"This task was a special experience. A real exercise against falling and feeling what happens when you will fall"* (GR assessment)

Overall, the participants reported the first RGR task as less fun and more intense because they were more scared to fall. As the RGR tasks progressed, participants generally started to report the tasks becoming more challenging and having to be focused when the AST was added. One participant noted:

*"I did not like the task, because I was startled every time"* (RGR AP)

Regarding task difficulty, our quantitative criteria were met. No participants used harness assistance to complete the task or had to stop due to instability or fatigue. Therefore, the feasibility criterion for task difficulty was fulfilled, despite some participants finding some tasks challenging.

All numerical scores related to the feasibility are shown in Fig 5 and all participant answers to the open questions per task are shown in the supplementary material.

## Discussion

This pilot study addressed the feasibility and multiple design considerations for technology-assisted fall-resisting skills training trials in older adults. All predefined feasibility criteria related to aim five were met, indicating that the fall-resisting skill tasks are feasible for older adults. The results regarding aim 1–4 are discussed in detail below, including limitations and implications of our results for future research and interventions, as well as for our own upcoming randomised controlled trial of fall-resisting skills training in older adults.

The results related to our first aim indicated that the three non-task-specific balance tasks (boat, city ride, and walking with AST) combined with each other can be used as a placebo in future RCT training fall-resisting skills, since almost all participants believed that these tasks together as a training would improve balance. For the individual tasks, we found that the standing weight-shifting tasks were perceived as actual balance training, suggesting that these tasks may be suitable as placebo tasks for fall-resisting skills trials. Walking with a dual task was less perceived as balance training, compared to the weight shifting tasks, as might be expected. For our upcoming RCT, this task will still be included to ensure the placebo control group complete a comparable amount of time walking and performing the AST as the fall-resisting skills

training groups in order to match the physical and cognitive load of the training. To be successful as a placebo control intervention, progression in task difficulty within but especially between consecutive training sessions will need to be incorporated [40].

For our second aim, we hypothesized that the larger obstacles, shorter available response times, and faster approach speeds would lead to increased obstacle crossing challenge. Smaller obstacles with longer available response times moving at the speed of the treadmill were expected to be the easiest combination. Our results showed that no condition caused a floor- or ceiling effect, suggesting that these combinations of obstacle parameters were suitable for the older adults and that this range of parameters should be feasible for future trials. Regarding our hypotheses, these could not be conclusively supported, since the differences in obstacle hits between the different obstacle parameter combinations were small and inconsistent. Despite this, we did find a significant difference between the easiest (medium obstacle size with faster approach speed and longer available response time) and hardest (larger obstacle sizes with faster approach speeds and longer available response time) conditions. Interestingly, the available response time did not significantly influence performance in the most difficult conditions. This contrasts with previous studies, which reported that shorter available response times generally impaired obstacle avoidance performance [24,65–67]. This difference may be because those studies used much shorter available response times (200–450 ms) than we did (1–2 strides: ±1000–2000 ms). In general, we might have been able to detect clearer differences between conditions had more than six trials per condition been included, which could be investigated in the future.

The third aim of this study was to explore evaluation methods for determining a stability loss threshold for GR assessment. The 3SD MoS threshold did not reveal the expected overall pattern of more participants reaching the threshold as the percentage increase in belt speed during the perturbation was raised. In hindsight, this may be, in part, because the first recovery step might not capture the complete reactive gait recovery response. Previous studies using the same setup and perturbation method have shown that deviations from unperturbed MoS do not necessarily occur only on the first post-perturbation step [30,51]. Therefore, future research could consider a multi-criterion threshold to improve the accuracy and reliability of the approach. For example, we could expand the number of steps included in the MoS determined threshold to two or three to capture all possible types of responses to the perturbation, as described above. Depending on the success of this approach, additional criteria, such as requiring at least two steps outside of the 3SD limit could be considered. In the current study, the restriction on only the first recovery step might explain why this method did not completely align with the researcher observations and ratings, since the researchers were not limited to evaluating only the first step after each perturbation (and the researchers' observations were generally consistent with each other). Regarding the participants' perceptions, these did not align well with either the MoS threshold or with the researchers' observations. Additionally, there was not perfect alignment between the participants responses to the different questions. These findings agree with previous research of Kapur, et al. [68] who similarly found that older adults often have difficulty accurately judging their own balance. For future trials, the implication of these findings is that older adults may not give objectively correct answers when asked about their stability and therefore cannot be relied on as a method for evaluating or progressing the difficulty of assessment or training tasks, despite these responses providing useful information regarding their balance confidence or falls efficacy. One general consideration of this GR task is that with the repeating perturbations, there may be an inherent learning effect that benefits participants. The implication of this is that the outcome of the test includes not only the absolute GR of the participant, but also their ability to learn and improve their responses within the test. This is unavoidable but should be kept in mind when interpreting the outcomes.

For the fourth aim, we hypothesised that with increasing unpredictability, anxiety levels would increase. However, our results suggested that with increasing unpredictability, anxiety levels decreased. A possible explanation is that the repeatedly experienced perturbations made the participants less anxious because they became more familiar with the overall experience. This aligns with Gerards, et al. [69] and Gerards, et al. [38], who suggested that some individuals may benefit from a more gradual progression of perturbations to help them build confidence and reduce anxiety. Our findings

align with the study by Wong, et al. [37] who also found no differences in perceived stress between groups with different intensity schedules during stance perturbations. In contrast, our findings differ from Okubo, et al. [36], who reported greater anxiety with more unpredictable perturbations. However, their setup used an overground walkway with slipping tiles and tripping boards, possibly creating a more intense experience than our treadmill- and platform-based perturbations. This suggests that anxiety responses could depend on the specific perturbation setup. Future research should include pilot testing of the specific perturbation setup to check its effect on anxiety. The relatively low anxiety scores after the first trial could be explained by the order of our protocol. Prior to the RGR tasks involving large perturbations, participants first completed the GR tasks, which included walking with smaller perturbations. This prior exposure to less intense perturbations may have caused the participants to get used to the perturbations and reduce their perceived anxiety in the following, more challenging perturbations. Even though the Hasselt protocol introduced the dual task earlier after fewer perturbations, anxiety scores remained relatively low in both groups. This suggests that repetition plays an important role in reducing anxiety during perturbation tasks.

A limitation of this study is that all tasks were performed in one session, instead of a complete training programme. While this approach allowed us to efficiently explore feasibility and task design, it limits the conclusions about training-related perceptions and improvements. In addition, participants' perception of improvement was assessed for the non-task-specific balance tasks only, not for all tasks, and measured only after a single session. Technical errors during data collection led to some missing data, which may have influenced the completeness and reliability of certain assessments. In part, these errors could later be corrected by altering the scripts in the applications and by addressing a memory issue related to saving the video files to ensure that these would not occur again in the trial. Finally, readers should take care in applying or extrapolating these results to the assessment, training or trials in other populations, since the current pilot study and our own related trial focus explicitly on healthy, community-dwelling older adults.

## Conclusion

Weight-shifting tasks are perceived as balance training by most older adults, indicating their potential as placebo tasks for fall-resisting skills trials. In our proactive gait adaptability task, the combination of large obstacle size and fast obstacle approach speed affected task difficulty the most. A single 3SD threshold based on the MoS may not be sufficient as a measure of GR since this is sometimes inconsistent with gait researchers' observations and participants' perceptions of stability. Using increasing types of perturbations and dual tasks does not increase anxiety, given they are introduced in this order (i.e., repetition reduces anxiety). Overall, the fall-resisting skill tasks used in this study are feasible for older adults.

## Supporting information

**S1 File. Supplementary Materials.** Supplementary file containing missing data table and questionnaire responses. (PDF)

## Author contributions

**Conceptualization:** Elisabeth G. van der Hulst, Kenneth Meijer, Pieter Meyns, Christopher McCrum.

**Data curation:** Elisabeth G. van der Hulst.

**Formal analysis:** Elisabeth G. van der Hulst.

**Funding acquisition:** Kenneth Meijer, Pieter Meyns, Christopher McCrum.

**Investigation:** Elisabeth G. van der Hulst.

**Methodology:** Elisabeth G. van der Hulst, Pieter Meyns, Christopher McCrum.

**Project administration:** Elisabeth G. van der Hulst, Kenneth Meijer, Pieter Meyns, Christopher McCrum.

**Resources:** Kenneth Meijer, Pieter Meyns, Christopher McCrum.

**Software:** Elisabeth G. van der Hulst.

**Supervision:** Kenneth Meijer, Pieter Meyns, Christopher McCrum.

**Visualization:** Elisabeth G. van der Hulst.

**Writing – original draft:** Elisabeth G. van der Hulst, Pieter Meyns, Christopher McCrum.

**Writing – review & editing:** Elisabeth G. van der Hulst, Kenneth Meijer, Pieter Meyns, Christopher McCrum.

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
