## [Decision Letter · Decision Letter 0]

4 Jan 2026

PONE-D-25-55551Design Considerations for Technology-assisted Fall-Resisting Skills Training Trials in Older Adults: A Pilot and Feasibility StudyPLOS One

Dear Dr. McCrum,

Thank you for submitting your manuscript to PLOS ONE. After careful consideration, we feel that it has merit but does not fully meet PLOS ONE’s publication criteria as it currently stands. Therefore, we invite you to submit a revised version of the manuscript that addresses the points raised during the review process.

We look forward to receiving your revised manuscript.

Kind regards,

Ryota Sakurai, Ph.D.

Academic Editor

PLOS One

Journal Requirements:

This work was supported by the special research fund 2022 call for doctoral grants in the framework of BOF UHasselt – Maastricht University cooperation [BOF22DOCUM13].

Additional Editor Comments :

The two reviewers have raised several important points. Please review all comments carefully and provide detailed responses.

Reviewers' comments:

Reviewer's Responses to Questions

**Comments to the Author**

1. Is the manuscript technically sound, and do the data support the conclusions?

Reviewer #1: Yes

Reviewer #2: Partly

2. Has the statistical analysis been performed appropriately and rigorously?

Reviewer #1: Yes

Reviewer #2: Yes

3. Have the authors made all data underlying the findings in their manuscript fully available?

Reviewer #1: Yes

Reviewer #2: Yes

4. Is the manuscript presented in an intelligible fashion and written in standard English?

Reviewer #1: Yes

Reviewer #2: Yes

5. Review Comments to the Author

Reviewer #1: Thank you very much for this well written manuscript which gives a thorough background information on current topics in falls prevention as well as very helpful insights for the design of future training trials.

I have some minor concerns for you to consider:

A small general remark: to my knowledge sentences like “… particularly in healthy older adults who do not necessarily yet feel, recognise or admit the need to participate in fall prevention interventions“ require a comma before the “or“ (bananas, peaches, or apples). Not being a native speaker myself, I have no strong feelings about this but maybe check. This would apply to several sentences in the text.

Introduction:

As community-dwelling older people are a very heterogenous group it might be helpful if you describe more precisely whether you are focusing exclusively on healthy, independently living older people without frailty, cognitive impairment, or a history of falls (primary prevention). This appears to apply to the sample of your pilot study. However, it would be helpful to know which of the citations you used in the introduction refer to this group. Additionally, it would be interesting if you focus solely on this group for your planned RCT.

p.4. second paragraph: “For reactive balance during stance perturbations, the Stepping Threshold Test 27,28 is an example of a progressive, threshold based test that can analyse robustness of standing stability to perturbations of increasing intensity but an equivalent for GR that could objectively determine when a person deviates from “steady state gait” due to overt stability loss does not yet exist, though a dynamic version of the Stepping Threshold Test using subjective judgement of stability loss has been reported 29.“

This is a very long and complex sentence describing the important distinction between static reactive balance and dynamic reactive balance. Assessments of dynamic reactive balance control is difficult and so far, we do not have satisfying solutions. Please make at least two sentences of this important aspect.

Methods

Experimental set-up: The sentence “For two tasks (boat, city ride), these six markers were covered and two shoulder markers were added to control the application.“ Is not very clear for the reader at that point. Although it is self-explicatory with further reading it might be helpful to rephrase, e.g: “For the two weight-shifting tasks (visual boat and car ride), these six

markers were covered and two shoulder markers were added to control the application.

Experimental procedure:

I think it very interesting and important that you used the participant’s stability -normalized walking speed rather than preferred speed (maybe explain why). However, please make the sentence clearer to understand: “These trials were recorded and during a rest

period following the familiarisation, participants’ stability-normalised walking speed 46 was calculated (see

calculation procedure in “Data processing” section below) and used for all following walking tasks to

ensure that all participants walked at approximately similar stability during all tasks.“ Make it clearer, that you calculated each participant’s individual stability-normalized walking speed. Possibly you could already give a hint how this was achieved at this point, e.g.: “These trials were recorded and a rest period following the familiarisation was used to calculate the individual stability-normalised walking speed (resulting in a MoS of 0.05m, see calculation procedure in “Data processing” section below)46 for each participant which was used for all following walking tasks to ensure that all participants walked at approximately similar stability during all tasks.

p.7: obstacle avoidance test: please clarify whether and how it was measured whether the participant successfully avoided the obstacle. What happened if the participant stepped on or touched the virtual obstacle? Was there a perturbation or any form of feed-back?

Is this obstacle avoidance task not planned for the RCT? Or will it be adapted according to the data of the pilot trial? (I ask, as you explicitly point out, that the stepping stone task will be used in the RCT.)

p.7: subjective feeling of stability: Please explain why you used a 7-point scale (a 10 point numerical rating scale is used for many applications e.g. pain). Or is it a Lickert scale?

table 1: it would be helpful to the reader to repeat the aims in an abridged form in the legend of the table: e.g: 1) are non-task-specific balance tasks potential control tasks; 2) obstacle parameters affect difficulty; 3) determining a stability loss threshold for GR assessment; 4) effect of perturbation task unpredictability on anxiety; and 5) feasibility of various fall-resisting skill tasks

p. 11, second paragraph: This sentence lacks a noun, I believe: “The

descriptively and visually compared the outcomes of the three assessment methods (MoS, participant

perception, and researcher evaluation), as well as the agreement between the gait researchers.“

Results

p.12: “11 older adults were recruited in this study.“ Avoid starting a sentence with a digit.

“with a mean of 2 falls.“ Better: with a mean of two falls. Avoid digits for numbers <10.

Missing data: please outline how you are going to avoid missing data in the planned trial. How many complete data sets did you gather? If I understand correctly 1 dropped out in the obstacle test due to knee pain, 1 only first half of task due to technical issues, 1 GR test stopped, 1 poor data quality, 3 no video data. I cannot tell if these are all different participants. If this were the case that would mean 7 incomplete data sets out of 11? Please describe your troubleshooting.

p.16: consider rephrasing this sentence “However, as the percentage

increase in belt speed during the perturbation increased, some discrepancies occurred.“ (perhaps: However, as the percentage increase in belt speed during the perturbation was raised, some discrepancies occurred)

“These did not occur at the highest intensities, where both researchers consistently identified visible gait adjustments and deviations from normal gait, but rather at moderate intensities, where it was sometimes unclear

whether a recovery step had occurred.“ Is this not what one would expect? You do not discuss this in the discussion.

p.19, 5th line from the bottom: “something was happening but couldn’t place exactly what.“ Please replace couldn’t with could not.

p.19, 3rd line from the bottom and last line: “5 participants mentioned they felt a bit nervous or anxious after the first RGR task with AP perturbations.“ “While the other 6 participants did not report any nervousness, anxiety or stress during this task.“ Please do not use digits for numbers < 10. Please check the whole text for this, also 3rd and 5th line of page 20.

Figure 5: in the legend correct please: RGR: reactive gait recovery (no capital letters)

Discussion

Page 21, second paragraph: Consider rephrasing ”for our first aim, …”. Maybe this would be clearer: “The tests we conducted for our fist aim showed that the three non-task-specific balance tasks …”

Page 22, second paragraph: “… the percentage increase in belt speed during the perturbation

increased.“ to avoid using the word increase twice maybe rephrase: “… the percentage increase in belt speed during the perturbation was raised.“

Page 22, second paragraph: “Previous studies using the same setup and perturbation method have shown that deviations from unperturbed MoS do not necessarily occur only on the first post perturbation step 30,51 Therefore, future research could consider a multi-criterion threshold to improve the accuracy and reliability of the approach. For example, we could expand the number of steps included in the MoS determined threshold to two or three to capture all possible types of responses to the perturbation, as described above“.

To support this point: In a study comparing young adults with older adults with or without a history of falls Hackbarth et al. (Experimental Gerontology 2025) showed that whereas in young adults step width and length normalize after only 1 or 2 steps in the older adults groups step width stayed elevated beyond step 8 after perturbation.

Page 22 , last paragraph: “Regarding the participants’ perceptions, these did not align well with either the MoS threshold or with the researchers’ observations. Additionally, there was not perfect alignment between the participants responses to the different questions. These findings align with previous research of Kapur, et al. 68 who similarly found that older adults often have difficulty accurately judging their own balance.“

Consider replacing the word “align“ by another expression in one of the sentences.

Page 22, last paragraph: “For future trials, the implication of these findings is that older adults may not give objectively correct answers when asked about their stability and therefore cannot be relied on as a method for evaluating or progressing the difficulty of assessment or training tasks“ – I am not sure I can agree with this statement. It implies that there is one correct estimation of balance control ant that the people themselves cannot assess it correctly. Maybe we should consider two separate constructs, one the objective competence in reactive dynamic balance (for which we do not yet have a good measure) and the other the subjectively rated feeling of being in control of the person, which would be a concept of self-efficacy. We also know the objective fall risk and self-rated falls efficacy (e.g. measured with the FES-I) do not necessarily align but are constructs that interact.

Page 23, first paragraph: “One general consideration of this GR task is that with the repeating perturbations, there may be an inherent learning effect that benefits participants.“ As there are several PBT trials that show that rate of laboratory falls decreases within the training, consider phrasing this sentence stronger, e.g.: … an inherent training effect seems probable …”

The implication of this is that the outcome of the test includes not only the absolute GR of the participant, but

also their ability to learn and improve their responses within the test. This is unavoidable but should be

kept in mind when interpreting the outcomes.“ – I totally agree.

Page 23, second paragraph: “For the fourth aim, our results suggested, in contrast to our hypothesis, that with increasing unpredictability, anxiety levels decreased.“- it would be easier for the reader if you made two sentences, e.g.: “For the fourth aim, we hypothesized that with increasing unpredictability, anxiety levels might increase. However, anxiety levels decreased as over the course of the training perturbations became more unpredictable.“

Page 23, limitations: please include the selected study population as a limitation. Your results do not necessarily transfer to frailer, fall prone populations.

Reviewer #2: This manuscript reports a pilot study that examined the feasibility of, and several key design considerations for, technology-assisted fall-resisting skills training trials in older adults. Prior to publication, several points should be clarified. My comments are as follows.

1. Gait Robustness (GR) and Reactive Gait Recovery (RGR) may influence each other. Was it possible in this study to evaluate these two constructs separately? Please clarify how GR and RGR were operationally distinguished in the study design and analyses.

2. In the GR test, what instructions were given to participants regarding how to respond to the perturbations? In addition, what was the rationale for separating the GR task from the RGR task? Could reactive balance ability have been evaluated within the GR task itself? Please explain why the authors considered it necessary to evaluate GR and RGR using distinct tasks.

3. In the GR test, the perturbation direction appears to mimic a trip. If so, the margin of stability (MoS) quantified here may reflect only balance loss in the forward direction. Is it necessary to also consider backward balance loss (e.g., slip-like perturbations), which would challenge stability in the opposite direction? Please justify the focus on the current perturbation direction and clarify the scope/limitations of MoS interpretation in this context.

4. In Fig. 2, what does a negative MoS indicate? Does it represent balance loss in the forward direction, in the backward direction, or does it indicate loss of stability in the anteroposterior (AP) direction? Please provide a clear explanation of the sign convention and how it should be interpreted.

5. The manuscript states: “MoS values are normalised in SD based on MoS during 10 steps of normal walking before each perturbation.” Why were MoS values normalized by the standard deviation (SD)? Please provide the rationale with supporting references for this normalization approach.

6. Previous studies have suggested that balance control in the mediolateral (ML) direction is more challenging than in the anteroposterior (AP) direction, and that older adults have particular difficulty in controlling ML balance. In the present study, the evaluation appears to focus on AP stability only. Please discuss the implications of not assessing ML stability and consider adding brief discussion of ML-related outcomes/limitations in light of prior literature (e.g., the references below).

https://doi.org/10.2522/ptj.20090125

https://doi.org/10.1016/j.apmr.2008.01.023

https://doi.org/10.1093/ageing/afl078

https://doi.org/10.1038/s41598-022-18382-7

6. PLOS authors have the option to publish the peer review history of their article (what does this mean?). If published, this will include your full peer review and any attached files.

Reviewer #1: No

Reviewer #2: No

---

## [Author Response · Author response to Decision Letter 1]

24 Feb 2026

Please see "Response to Reviewers" file.

---

## [Decision Letter · Decision Letter 1]

10 Mar 2026

Design considerations for technology-assisted fall-resisting skills training trials in older adults: A pilot and feasibility study

PONE-D-25-55551R1

Dear Dr. McCrum,

We’re pleased to inform you that your manuscript has been judged scientifically suitable for publication and will be formally accepted for publication once it meets all outstanding technical requirements.

Kind regards,

Ryota Sakurai, Ph.D.

Academic Editor

PLOS One

Additional Editor Comments (optional):

The manuscript improved with comments from the reviewers. Thanks for providing important information.

Reviewers' comments:

Reviewer's Responses to Questions

**Comments to the Author**

1. If the authors have adequately addressed your comments raised in a previous round of review and you feel that this manuscript is now acceptable for publication, you may indicate that here to bypass the “Comments to the Author” section, enter your conflict of interest statement in the “Confidential to Editor” section, and submit your "Accept" recommendation.

Reviewer #1: All comments have been addressed

Reviewer #2: All comments have been addressed

2. Is the manuscript technically sound, and do the data support the conclusions?

Reviewer #1: Yes

Reviewer #2: Yes

3. Has the statistical analysis been performed appropriately and rigorously?

Reviewer #1: Yes

Reviewer #2: Yes

4. Have the authors made all data underlying the findings in their manuscript fully available?

Reviewer #1: Yes

Reviewer #2: Yes

5. Is the manuscript presented in an intelligible fashion and written in standard English?

Reviewer #1: Yes

Reviewer #2: Yes

6. Review Comments to the Author

Reviewer #1: All my concerns have been met. Thank you for the thourough revision and explanations. I am looking forward to the results of the planned RCT!

Reviewer #2: Thank you for your revisions. I believe the authors have appropriately addressed my comments and revised the manuscript accordingly.

7. PLOS authors have the option to publish the peer review history of their article (what does this mean?). If published, this will include your full peer review and any attached files.

Reviewer #1: No

Reviewer #2: No

---

## [Editor Report · Acceptance letter]

PONE-D-25-55551R1

PLOS One

Dear Dr. McCrum,

I'm pleased to inform you that your manuscript has been deemed suitable for publication in PLOS One. Congratulations! Your manuscript is now being handed over to our production team.

Kind regards,

on behalf of

Dr. Ryota Sakurai

Academic Editor

PLOS One